# Comparison of the Chemical Properties of Vinegar Obtained via One-Step Fermentation and Sequential Fermentation from Dragon Fruit and Pineapple

Nanthavut Niyomvong [1,2], Rachcha Sritawan [3], Jureeporn Keabpimai [3], Chanaporn Trakunjae [3,4] and Antika Boondaeng [3,*]

1   Department of Biology and Biotechnology, Faculty of Science and Technology,
    Nakhon Sawan Rajabhat University, Nakhon Sawan 60000, Thailand
2   Science Center, Nakhon Sawan Rajabhat University, Nakhon Sawan 60000, Thailand
3   Kasetsart Agricultural and Agro-Industrial Product Improvement Institute, Kasetsart University,
    Bangkok 10900, Thailand
4   School of Biological Sciences, University Sains Malaysia, Penang 11800, Malaysia
*   Correspondence: aapakb@ku.ac.th

**Abstract:** Dragon fruit has many potential health benefits. It is inexpensive and widely cultivated in Thailand. The addition of dragon fruit to pineapple vinegar may help enhance the total phenolic compounds and antioxidant activity. This study aimed to study and compare the chemical characteristics of vinegar produced via one-step fermentation of a mixture of pineapple and dragon fruit juice from Krok Phra District of Thailand using *Saccharomyces cerevisiae* var. *burgundy* with that obtained using sequential fermentation using *Saccharomyces cerevisiae* var. *burgundy* and *Acetobacter aceti.* When the two fermentation methods were compared on day 20, the maximum acetic acid concentration obtained from sequential fermentation was 5.79 ± 0.25%, which was higher than that obtained in one-step fermentation (1.93%). The total phenolic compound content in the mixed fruit vinegar obtained from sequential fermentation and one-step fermentation was 228.01 and 242.2 mg/L gallic acid equivalents, respectively. The antioxidant content of the products obtained in sequential and one-step fermentations was 187.91 mg/L GAE and 209.33 µg/g of Trolox equivalents, respectively, which was consistent with the total phenolic compound content. This indicated that the acetic acid content in the mixed pineapple and dragon fruit juice vinegar obtained using sequential fermentation was higher than that obtained using one-step fermentation although its total phenolic content and the antioxidant activities were slightly lower. These observations will be useful for improving vinegar fermentation in the area.

**Keywords:** dragon fruit; vinegar fermentation; one-step fermentation; sequential fermentation; antioxidants; total phenolics

## 1. Introduction

Fermented vinegar, a beverage that originated in European countries [1], is now used as a popular condiment and beverage worldwide for its good taste and nutritional value [2]. In general, vinegar, which is produced using fermented fruit, water, and microorganisms, where the raw materials are transformed into acetic acid, has a sour taste [3,4] and typically contains about 5% acetic acid by volume [4]. The color of fermented vinegar is usually brown, and it contains a gelatinous disc or "vinegar mother" [5]. Typically, before packaging or consumption, the products are not heat-treated, and because of this, live microorganisms can still be found in beverages; hence, these drinks are classified as probiotic drinks [6].

Fermented vinegar is originally made from apple juice, which is sweet, has good taste, and worldwide consumption and production [7]; however, apples are more expensive than the local Thai fruits because they cannot grow in all regions of Thailand. Hence, attempts

should be made to use local fruits that grow abundantly in fermenting vinegar. For example, dragon fruit and pineapple, which are both inexpensive and found abundantly in the Krok Phra District of Nakhon Sawan Province, the Ban Rai District of Uthai Thani Province, and most of the other regions of Thailand, can be used as raw materials for producing fermented vinegar.

Dragon fruit, a member of the family Cactaceae native to the tropical regions of North, Central, and South America [8], is now widely cultivated in South East Asia; it is of two types: the first type is *Hylocereus undatus* (Haw.), which is a pink-skinned fruit with white flesh [9], and the second type is *H. polyrhizus* (Weber) Britt. & Rose, which is a red-skinned fruit with red flesh. In 1991, dragon fruit was widely cultivated in Thailand [10]; although it tasted bad in the beginning, the cultivar was later modified to improve the taste. Fresh dragon fruit is commonly used as an ingredient in food and drink. Dragon fruit showed beneficial biological activities against pathogenic microbes and diseases such as diabetes, obesity, hyperlipidemia, and cancer [11]. It not only enhances health by biological effects, but is also a low-calorie food (about 50–60 kcal/100 g) high in fiber content that contains various vitamins and minerals as well as phytonutrients named betalains [12]. Betalains, which are also found in the red peel and flesh, possess antioxidant, anti-inflammatory, and detoxification properties that are beneficial for the human body [13–16]. In the industry, betalains are used to make food coloring, as they are highly safe for consumption [16]. Furthermore, the flesh of dragon fruit contains vitamin C, which stimulates the immune system, and high levels of potassium, which improves and maintains bone structure [17]. The Department of Agricultural Extension of Thailand [18] surveyed dragon fruit planting areas in all regions of Thailand in 2019 and found that the planting area covered 11,329 acres and that the total yield was 42,569 tons per year [18], which is sufficient for it to be used as a raw material in fermented vinegar production. Additionally, in the Nakhon Sawan Province and nearby areas, dragon fruit is grown in backyard gardens, harvested, and sold in the market. A study showed that the sugar content of the raw material can directly affect the acetic acid content of the final vinegar [19]; hence, dragon fruit products, which possess good nutritional properties, can be used together with pineapple, another fruit found abundantly in Thailand, for fermentation. A recent survey showed that 1.96 million tons of crops are harvested annually from 192,675 acres of pineapple plantations [20], which is considered a high production volume at low cost. This renders pineapple suitable for use as a low-cost raw material for producing high-quality fermented vinegar; furthermore, pineapple is a nutritious and tasty fruit with high vitamin C, sugar, and acid content and a specific aroma, which makes it a suitable raw material for producing various beverages.

Vinegar was prepared using various methods to transform alcohol into acetic acid in a short period. The one-step fermentation process was introduced with the advantages of low operating cost and high yield of nutritional substances, such as amino acids, organic acids, and volatile compounds [21], which give a unique taste to vinegar. However, one-step fermentation must take into account various parameters, such as pH, temperature, oxygen concentration, and substrate [22]. The sequential fermentation process consisted of two steps: alcoholic and vinegar fermentation. Vinegar fermentation through the surface culture fermentation (SCF) process [23] is a simple method to provide a high acetic acid yield in a short time.

The goal of this study is to find out more about the chemical properties of local fruit vinegar that has been fermented using monocultures and mixed cultures. Moreover, to find the best conditions for vinegar fermentation, a comparison was also made between one-step and sequential fermentation of fruit vinegar.

## 2. Materials and Methods

### 2.1. Culture Preparation

*Saccharomyces cerevisiae* var. *burgundy*, the yeast commonly used for wine fermentation, was obtained from the Department of Applied Microbiology, Institute of Food Research and Product Development (IFRPD), Kasetsart University, Thailand. It was grown on YPD

agar (10 g/L yeast extract, 20 g/L peptone, and 18 g/L agar, supplemented with 20 g/L glucose) at 30 °C for 24–48 h and was used as the inoculum. The *Acetobacter aceti* TISTR 354 starter culture, provided by the Thailand Institute of Scientific and Technological Research (TISTR), was maintained on GYC agar (10 g/L yeast extract, 20 g/L CaCO$_3$, and 15 g/L agar, supplemented with 50 g/L glucose) [24].

### 2.2. Fruits

Ripe dragon fruit and pineapple were harvested from agricultural sources in Krok Phra District of Nakhon Sawan Province, Thailand. Matured and undamaged fruits of similar size that had undamaged skin were selected for production.

### 2.3. One-Step Fermentation and Sequential Fermentation Using S. cerevisiae var. burgundy and A. aceti

#### 2.3.1. Preparation of Juices

Fruit samples at a ripe stage were cleaned with tap water, peeled, and freshly crushed. Pineapple juice and red dragon fruit juice and water were mixed in a 1:0.5:1.5 ratio and used to fill the container. The TSS and pH of the mixed fruit juices was adjusted to 25 °Brix by adding sucrose and four by adding baking soda or citric acid, respectively, and used for setting up the starter culture. Thereafter, the mixed juice was sterilized using 150−200 mg/L potassium metabisulphite (K$_2$S$_2$O$_5$) and stored overnight at 30 °C for further experimentation.

#### 2.3.2. Preparation of Starter Cultures

The mixed TSS (25 °Brix) and pH (4) -adjusted fruit juice was used for setting up the starter culture. The 24-h-old *S. cerevisiae* var. *burgundy* yeast culture was transferred to 100 mL of sterilized mixed fruit juice in a 500 mL bottle, stored at 30 °C for 24 h, and used as the starter culture.

To prepare the starter culture of *A. aceti* TISTR 354, 90 mL of sterilized mixed fruit juice, with TSS adjusted to 5 °Brix by adding sterilized water, was mixed with 3 mL of 95% ethanol and 7 mL of *A. aceti* (*v/v*, 10$^6$ CFU/mL) [23] and incubated at 30 °C for 72 h before use.

Vinegar was fermented from mixed fruit juice in a 5 L food-grade plastic bottle; the bottle was filled with 1 L of the mixed fruit juice with adjusted pH and TSS, to which the starter culture was added, and fermentation was performed as mentioned below:

1. One-step fermentation: Mixed starter cultures were added at the beginning of the fermentation in 1:1 ratio and fermented at 30 °C for 20 days. The experiment was repeated thrice, and the samples were collected to analyze the composition of the fermented product;

2. Sequential fermentation: This method was performed using the SCF process [23], which consists of two steps. *S. cerevisiae* var. *burgundy* was added at the start of the fermentation, which was performed at 30 °C for 10 days, following which the yeast was inactivated by adding 150–200 mg/L KMS. Vinegar fermentation was performed by mixing sterilized juice mixture, mixed juice wine, and starter culture of *A. aceti* TISTR 354 in the ratio of 600:300:100. After 2 days of incubation at 30 °C, 1000 mL of the mixed fruit wine was added to the container and incubated for 18 days. The samples were collected after every 24 h to analyze the composition of the fermented product. The experiment was conducted in triplicate.

### 2.4. Analysis of the Composition of the Fermentation Product

#### 2.4.1. Analysis of Alcohol (Ethanol) Content by Volume (Degrees)

The ethanol concentration was performed using gas chromatography (Chromosorb-103, GC4000; GL Sciences; Tokyo, Japan) with an HP5 capillary (30 m × 0.32 mm × 0.25 μm; JW Scientific, CA, USA) and FID detector under the following conditions: split flow, 50 mL/min; air flow, 250 mL/min; N$_2$ carrier flow, 30 mL/min; column temperature,

185 °C; injector temperature, 250 °C; detector temperature, 250 °C. n-Propanol was used as the internal standard for comparison [25].

### 2.4.2. Analysis of Acetic Acid Content

A high-performance liquid chromatography column (HPLC, Shimadzu LC-20A, Japan) was connected to the Aminex HPX-87H column and UV detector using 5 mM $H_2SO_4$ as the mobile phase under the following conditions: flow rate, 0.6 mL/min; column temperature, 60 °C; wavelength, 210 nm [26].

### 2.4.3. Analysis of Reducing Sugar Content Using the Dinitro Salicylic Acid (DNS) Method

The reducing sugar content was determined using the DNS method [27]. The sample of 0.5 mL was mixed with 0.5 mL of DNS solution and boiled for 10 min. The sample was cooled down by immersing the sample tube into cold water immediately. After that, 5 mL of water was added and mixed well. The absorbance was measured at 540 nm and compared with the standard curve using glucose standard solution.

### 2.4.4. Analysis of Total Soluble Solids

TSS was analyzed using a refractometer (RHB-32ATC, Shenzhen City, China), reported as °Brix for soluble solid contents.

### 2.4.5. Total Viable Plate Count

Next, 10 mL of vinegar samples was aspirated and mixed with 90 mL sterile normal saline (0.85%) using aseptic techniques and shaken well. The samples were diluted by pipetting 1 mL vinegar into 9 mL sterile distilled water, resulting in $10^{-1}$ fold to $10^{-4}$ fold dilutions of the sample; then, 0.1 mL of the different dilutions of the vinegar sample was plated on agar plates. *S. cerevisiae* var. *burgundy* was cultured on plate count agar (PCA) and *A. aceti* TISTR 354 on GYC agar; they were spread on agar plates using a sterile spreader, after which the plates were left to dry. The plates were incubated at 30 °C for 48 h. After incubation, the total number of colonies on the agar plates was counted and reported as the number of colonies per milliliter of the samples.

### 2.4.6. Determination of Total Phenolics Content

The content of the total phenolic compounds in the wine sample was determined using Folin–Ciocalteu colorimetric method [28]. Briefly, 0.3 mL of each sample was mixed with 1.5 mL of the Folin–Ciocalteu reagent, to which 1.2 mL of 7.5% (*w*/*v*) sodium carbonate solution was added. The samples were incubated in the dark for 30 min at room temperature. Percentage absorbance was measured using a spectrophotometer (Thermo Fisher Scientific 4001/4 Genesys 20, Waltham, MA, USA) at 765 nm. A calibration curve was established using a gallic acid standard solution.

### 2.4.7. Determination of Antioxidant Activity

The total antioxidant activity of vinegar was estimated using the DPPH radical scavenging capacity assay following the procedure described by Vidal-Gutiérrez et al. [29]. The scavenging activity of the free radicals was evaluated by measuring the absorbance at 517 nm after a 30 min reaction in the dark, using a UV–vis spectrophotometer (Shimadzu UVmini-1240, Kyoto, Japan). Results were expressed as µg equivalents of Trolox/g of fresh sample.

## 3. Results

### 3.1. Chemical Characteristics

The basic chemical characteristics of the pineapple and red dragon fruit juices are shown in Table 1. The TSS of pineapple juice of 13.07 °Brix was slightly higher than that of red dragon fruit juice (11.93 °Brix). The pH of pineapple juice was lower than that of red dragon fruit juice because of higher TTA. However, the nitrogen content of both juices

was >0.025 g/L *w/v*, which is sufficient for yeast growth [30]. The TPC and antioxidant activities determined by Folin–Ciocalteu reagent and the DPPH radical scav.n that of red dragon fruit juice. Owing to the total concentration, these juices could be used as a raw material for vinegar fermentation.

**Table 1.** Basic chemical characteristics of fresh juices.

| | Value ± SD | |
| --- | --- | --- |
| **Chemical Characteristics** | **Pineapple Juice** | **Dragon Fruit Juice** |
| pH | 3.58 ± 0.02 | 4.44 ± 0.01 |
| Total soluble solid (TSS, °Brix) | 13.07 ± 0.12 | 11.93 ± 0.23 |
| Total titratable acidity (TTA, as citric acid) (%*w/v*) | 0.286 ± 0.00 | 0.124 ± 0.00 |
| Nitrogen content (%*w/v*) | 0.08 ± 0.01 | 0.027 ± 0.20 |
| Antioxidant activity (μg TE/g) | 198.29 ± 1.51 | 162.60 ± 8.13 |
| Total phenolic compound (mg GAE/L) | 407.00 ± 8.60 | 256.50 ± 0.58 |

*3.2. One-Step Fermentation and Sequential Fermentation Using Saccharomyces cerevisiae* var. *burgundy and Acetobacter aceti*

Comparative study of one-step fermentation and sequential fermentation of pineapple and dragon fruit juice revealed the following:

When the mixed pineapple and dragon fruit juice was fermented to vinegar by adding *S. cerevisiae* var. *burgundy* and *A. aceti* TISTR 354 in the ratio of 1:1 (one step), we found that the acetic acid content increased slightly, and the highest value obtained was 1.93 ± 0.01% on day 20 (Figure 1a). Ethanol content increased rapidly from the first day to day 15, with the highest value of 6.98 ± 0.071%. However, acetic acid production was relatively low (1.93 ± 0.01%). Reducing sugar concentration decreased slightly from initial fermentation to day 6 and then increased to day 8. In the initial fermentation, yeast converts the reducing sugars present in the fruit juice into alcohol and carbon dioxide through the fermentation process. Then, they generate a mixture of fructose and glucose from sucrose, which results in the decrease of reducing sugar in the first stage of fermentation and then an increase (Figure 1a).

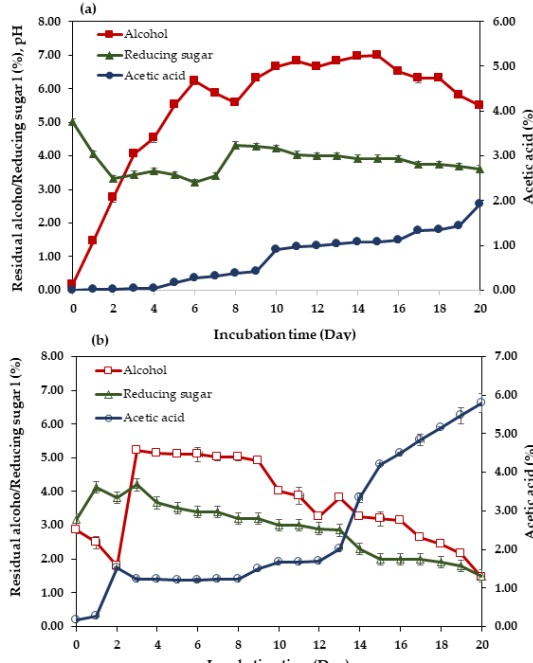

**Figure 1.** Changes in reducing sugar, acetic acid, and residual alcohol concentrations of fruit vinegar in one-step fermentation (**a**) and sequential fermentation (**b**).

Sequential fermentation is a two-stage process. The first step is the growth of *S. cerevisiae* var. *burgundy* until 10% ethanol is produced, following which the yeast is inactivated, and fermentation is continued to form fermented vinegar when 600:300:100 ratio of mixed fruit juice:mixed fruit wine:vinegar starter is used. After adding the mixed fruit juice wine at 48 h of fermentation, the acetic acid content increased by $1.52 \pm 0.02\%$ over 2 days (Figure 1b). Alcohol was oxidized by *A. aceti* TISTR 354 in the presence of air to acetic acid, which correlated with decrease in pH from 4.06 to 3.42. This is in accord with the results obtained by Sossou et al. [31], who reported the change of pH from 4.4 to 2.9 during vinegar fermentation from pineapple peeling and processing.

To increase the efficiency of acetic acid production, 1000 mL of the mixed fruit wine was added to the fermentation tank after 2 days; however, the acetic acid content decreased from $1.52 \pm 0.02\%$ to $1.23 \pm 0.01\%$, while that of ethanol increased from $1.80 \pm 0.01\%$ to $5.23 \pm 0.04\%$ (Figure 1b). The increase in ethanol content increased the acetic acid level, as *A. aceti* TISTR 354 oxidizes ethanol as the substrate to produce acetic acid. The results showed that acetic acid content increased from $0.18 \pm 0.01\%$ to $5.79 \pm 0.25\%$ in 20 days, while 1.46% alcohol remained (Figure 1b). This result was comparable to that reported by Roda et al. [32], which showed an increase of 0.05 to 5% ($w/v$) with a residual alcohol concentration of 0.5% during 30 days of pineapple vinegar fermentation, and Sossou et al. [31], in which acetic fermentation of pineapple peel produced vinegar at 4.5% ($w/v$) over 20 days. Raji et al. [33] studied pineapple vinegar fermentation from pineapple waste and observed an acidity value of 4.77% ($w/v$) after 11 days of acetic acid fermentation.

According to the experimental results, the acetic acid yield of the mixed pineapple and dragon fruit juice vinegar obtained from sequential fermentation was higher than that achieved in one-step fermentation. In this study, the sequential fermentation method was separated into two steps, in which each step required only one microorganism, which causes no growth inhibition among microorganisms. However, microorganisms' growth inhibition might occur in one-step fermentation due to the mixture of various microorganisms. Interestingly, adding mixed fruit wine on day 2 of the second step fermentation may have increased the volume of oxygen in the liquid media, which promoted the growth of *A. aceti* TISTR 354 and enhanced the conversion of alcohol to acetic acid [23]. Furthermore, variations in raw materials, the amount of acetic acid bacteria added, and fermentation time all influence the acidity level of vinegar products [34].

### 3.3. The pH of Vinegar Fermentation

Figure 2 shows that the reduction in pH correlated with the acetic acid content. The pH decreased from 4.05 to 3.58 in one-step fermentation and to 2.79 in sequential fermentation, which is indicative of higher acetic acid content. This result corresponds with those observed in previous studies. Roda et al. [32] studied pineapple vinegar fermentation from pineapple waste for 7–10 days during alcoholic fermentation using *S. cerevisiae* and for 30 days during vinegar fermentation using *A. aceti*. After 30 days of acetic acid fermentation, the pH of pineapple wine was 3.78 on day 1 and decreased to 3.00 on day 30. Singh and Singh [35] observed a decrease in the pH value of pineapple vinegar from 4.50 (initial pH) to 3.50 after nine days of fermentation. Sossou et al. [31] reported that during acetic acid fermentation, the pH of pineapple vinegar decreased from 4.40 to 2.90 on day 25.

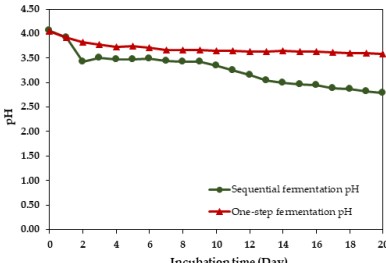

**Figure 2.** Change in pH of fruit vinegar in one-step and sequential fermentations.

### 3.4. Total Viable Cell Count

Analysis of the total culture in the one-step fermentation (Figure 3a) revealed that both the cultures grew well. The *S. cerevisiae* var. *burgundy* culture showed maximum increase on day 1 with $1.78 \times 10^7$ CFU/mL, which then decreased through the last day for four log cycles. The *A. aceti* TISTR 354 culture showed maximum increase on day 2 with $2.70 \times 10^7$ CFU/mL and decreased through the last day for four log cycles. Similarly, sequential fermentation showed an exponential phase of *A. aceti* from $1.46 \times 10^6$ on day 0 to $2.23 \times 10^7$ CFU/mL on day 1, a stationary phase from day 1 to day 2, and a decline phase beyond day 2 to the end of fermentation (Figure 3b).

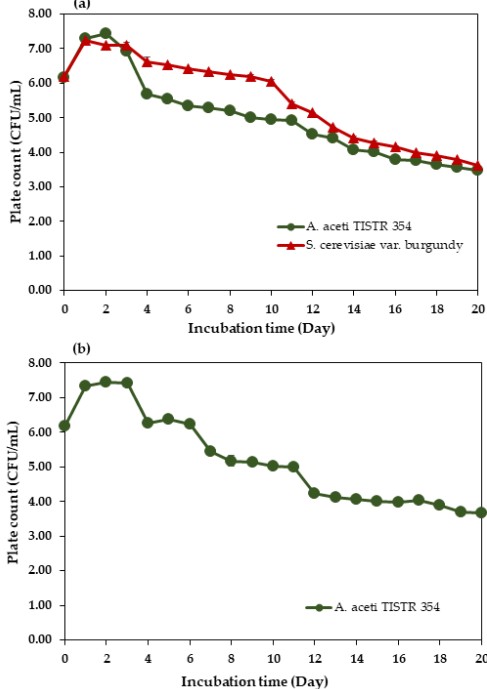

**Figure 3.** Change in microbial populations of fruit vinegar in one-step fermentation (**a**) and sequential fermentation (**b**).

### 3.5. Total Phenolic Content and Antioxidant Activity

In this study, the yield of acetic acid in sequential fermentation was higher than that obtained in one-step fermentation. On the last day of fermentation, the total phenolic compound content and antioxidant activities were determined using the DPPH assay. The values obtained for sequential fermentation and one-step fermentation were 228 mg/L gallic acid equivalents (GAE) and 187.91 µg/g of Trolox equivalents (TE) and 242.2 mg/L GAE and 209.33 µg/g TE, respectively (Table 2).

**Table 2.** Total phenolic acid and antioxidant activity of fruit vinegar during fermentation.

| Samples | TPC (mg GAE/L) | DPPH (µg TE/g) |
|---|---|---|
| The mixed fruit juice | 278.0 ± 4.32 | 215.57 ± 0.72 |
| The mixed fruit wine | 270.96 ± 10.91 | 156.88 ± 3.07 |
| Final vinegar from one-step fermentation | 242.2 ± 2.88 | 209.33 ± 0.97 |
| Final vinegar from sequential fermentation | 228.01 ± 3.62 | 187.91 ± 1.31 |

The content of the total phenolic compounds obtained in one-step and sequential fermentations tended to be stable from the first to the last day of fermentation. On the last day, the total phenolic content was 242.2 and 228.01 mg/L GAE for one-step and sequential fermentations, respectively, which were higher than that in pineapple vinegar reported

by Mohamad et al. [36] (169.67 mg/L GAE) and Boonsupa et al. [37] (128.96–172.34 mg/L GAE). This can be an advantage of using dragon fruit as the raw material for vinegar production, as phenolic compounds possessing antioxidant properties and can be found in many high-value plants, herbs, flowers, or fruits [38]. The phenolic group on the benzene ring in these compounds can act as a source of electrons or hydrogen, which can inhibit or slow down oxidation reactions [39].

In accordance with the announcement of the Ministry of Public Health of Thailand (No. 204) B.E. 2543 (2000), fermented vinegar should not contain <4% acetic acid and not more than 0.5% residual alcohol. The U.S. Food and Drug Administration (2007, FDA/ORA CPG 7109.22) [40] also stipulates that the acetic acid content in fermented vinegar should not be less than 4%. The European Union stipulates that fermented vinegar should contain at least 5% acetic acid and that the maximum residual alcohol content should be 0.5%. Wine vinegar obtained by fermenting acetic acid from wine should not contain <6% *w/v* acetic acid and not more than 1.5% v/g residual alcohol (Regulation (EC) No.1493/1999). According to our results, the composition of the vinegar obtained from mixed pineapple and dragon fruit juice in sequential fermentation was in agreement with the specifications of the U.S. Food and Drug Administration regarding acetic acid content in vinegar (Food and Drug Administration, 2007, FDA/ORA CPG 7109.22) [40]. In sequential fermentation, the acetic acid content with standard deviation showed $5.79 \pm 0.25$%, which is possibly higher than 6% or below. The residual alcohol is relatively high; therefore, the fermentation time could be extended to decrease the residual alcohol, which should not be more than 0.5%, and the acetic could reach more than 6%. However, acetic acid content was slightly low, and residual alcohol content was >1.5% in one-step fermentation, which is higher than the set limits; thus, fermentation time should be increased in this case to ensure complete oxidation of the alcohol to acetic acid. In addition, aeration may help the fermentation process promote cell growth and alcohol oxidation by acetic acid bacteria.

## 4. Conclusions

This study demonstrated that vinegar produced from mixed pineapple and dragon fruit juice via sequential fermentation contained higher levels of acetic acid than vinegar produced via one-step fermentation after 20 days of fermentation, which meets the standard acetic acid content; however, the residual alcohol exceeded 0.5%. Therefore, the fermentation time should be extended to enable oxidization of the alcohol to acetic acid. Additionally, enhanced oxygen supply may increase the acetic acid fermentation yield. The addition of dragon fruit juice to pineapple vinegar increased the total phenolic content. These findings will be useful for enhancing the production of vinegar using local fruit resources to create functional drinks with many benefits.

**Author Contributions:** Conceptualization, N.N. and A.B.; methodology, A.B.; software, N.N.; validation, R.S.; formal analysis, J.K.; investigation, C.T.; resources, A.B.; data curation, N.N.; writing—original draft preparation, N.N.; writing—review and editing, A.B.; visualization, N.N.; supervision, N.N.; project administration, N.N.; funding acquisition, N.N. All authors have read and agreed to the published version of the manuscript.

**Funding:** This research was funded by the Research and Development Institute, Nakhon Sawan Rajabhat University (RDI), Thailand, under the Fundamental Fund (FF) 2021, grant number FRB650025/0199.

**Institutional Review Board Statement:** Not applicable.

**Informed Consent Statement:** Not applicable.

**Data Availability Statement:** All data supporting the conclusions of this article are included in the manuscript.

**Acknowledgments:** The Science Center and Faculty of Science and Technology, Nakhon Sawan Rajabhat University, and Kasetsart Agricultural and Agro-Industrial Product Improvement Institute (KAPI), Bangkok, Thailand, provided support and facilities.

**Conflicts of Interest:** The authors declare no conflict of interest.

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
