# Peer review of "Comparison of the Chemical Properties of Vinegar Obtained via One-Step Fermentation and Sequential Fermentation from Dragon Fruit and Pineapple"

_beverages, doi:10.3390/beverages8040074_

Round 1

Reviewer 1 Report

These advises could be taken in consider for improving manuscript:

**More clear resolution is necessary for the figures.

*Also 1-2 sentenced could be added for results and discussions part for sugar content and its changes. 

Page 2, Line 83: Beginning from the ‘Vinegar was prepared using various ..’ can be divided into a new paragraph

Page 3, Line 110: Krok Phra District. (Kasetsart University, Thailand)

Page 3, Line 110-111: Sentence could be revise as ‘Matured and undamaged fruits of similar size that had undamaged skin were selected for production.’

Page 3, Line 118: Please add meaning of KMS?

Page 3, Line 120: Sentence could be revise as ‘The mixed TSS (25° Brix) and pH (4) adjusted fruit juice was used for setting up the starter culture.’

Page 4, Line 159: Please delete ‘(Brix Determination)’

Page 4, Line 160: Please add city and country for the ‘refractometer (RHB- 160 32ATC)’.

Page 4, Line 161: Please use ‘TSS’ for soluble solids.

Page 8, Line 298: Fermentation was produced 30 days, but in Table 1 only 2 days results are given: beginning and the 10th day. Why there is no result for 20th and 30th days??.. Please add relevant data. Or should it be Day 30??..

Reviewer 2 Report

The subject is exciting and innovative in dealing with the possibilities of the production and the chemical characteristics of vinegar produced via one-step fermentation of a mixture of pineapple and dragon fruit juice from Thailand using different microorganisms. The results may be helpful in the overall food chain in the global environmental changes and food scarcity in some parts of the world.

Author Response

Thank you for your comment.

Reviewer 3 Report

The article entitled "Comparison of the Chemical Properties of Vinegar Obtained via One-step Fermentation and Sequential Fermentation from Dragon Fruit and Pineapple" reports on the composition of two vinegars obtained with two fermentation methods on a mixed medium from pineapple and dragon fruit juices. The paper is well written, the aim is somewhat clear, and results description is sound. 
There are some key issues that, in mu opinion, prevent this study from being published. 

1) the fruit juices should have been characterised individually, especially regarding their phenolic content, before being mixed. Without this info, the discussion about phenolics and antioxidant potential attributed to dragon fruit is speculative. 

2) the materials and methods need to be better described

3) the description of the results does not match the data presented in figure 1

4) due to the high alcohol content, the obtained products cannot be called vinegar, and this is not stressed enough in the discussion and conclusions
Below are some specific comments:

L41-44: I am not sure that vinegars are bottled unfiltered, at least not in Europe. Please check this statement

L45-46: wine vinegar is dominant, at least in Europe. Please check this

L62: reference 11 refers to apple vinegar, while here the authors talk about dragon fruit benefits. Please check this

L64: a citation is needed after betalains

L69: a citaition is needed after excretion

L92-95: the aim needs to be rewritten as it is a bit too long and not hypothesis driven. Please rewrite it to improve clarity, and avoid the use of adjectives as appropriate, nutritious etc

Section 2.2. more info on the fruit would be appropriate, like when was it sourced, from were etc

Section 2.3.1. Details are missing, as: how was the juice obtained, how much sucrose was added? how was the pH adjusted? what is KMS?, at what temperature was the juice stored?

L124: how was the brix level adjusted to 5°?

L130: I do not find the description on how the mixed starter cultures have been prepared, please add it

L141: were sequential fermentation done in replicates? please add this info

section 2.4.1: please improve the description of the method, and add a reference if available

Section 2.4.2: citation of the method is missing

Section 2.4.3. more info needed (sections 2.4.6 and 2.4.7 are good examples on how the methods should be reported)

L195: ethanol content does not increase to the last day, it actually decreases from day 14. please rewrite this

Figure 1a: An explanation on why the sugar content increases at day 7 needs to be given

L205: citation 26 seems to be out of place, I would have expected the reference to figure 2 instead

L205-209: I am confused, where are these data shown? When was teh liter of of mixed fruit wine added to the fermentation tank? Please revise this section and explain better the procedures used.

L209-211: I do not find these data in the paper. From figure 1a at day 18 I see alcool levels of 2.1% (not 1.46%), and acetic acid level of 5.2% (not 5.79%). please check these data carefully

Figure 1b. In the text it should be better explained why the starting point in acetic acid is not 0 but close to 3%. I understand why, but it should be better explained.

L217-226: This section needs to be explained better as it is a bit unclear

L239-240: this is obvious, please delete this sentence

L279-280: also this relationship is expected

L281-290: are there information on hte phenolic content of the pineapple and dragon fruit juices alone before mixing? This would be a much more definitive way to understand the individual contribution of phenolic compounds

Table 1: there is a symbol missing (228.013.62)

L315: from my undestanding of figure 1, alcohol content is way higher than limits in both samples, and this should be stressed more in this section. This should also be stressed in the conclusion that, at the moment, are misleading.

Round 2

Reviewer 3 Report

The authors have accepted most of my previous suggestions and improved the manuscript. I think that with further minor revisions the manuscript could be better. Below are some additional comments that require attention:

Figures and tables should be better placed within the text

L251: "No" is repeated twice

To be precise, 5.79% acetic acid is below 6% even when accounting for the StDev. The 0.25% variation could bring it just over 6, but also just over 5.5%. Please stress this fact in the discussion and conclusions

L127: I undestand that water was used to reach 5% brix, but this info needs to be stated in the materials and methods

L235-241: add a refence to Figure 1a so that the readers know where to find the data mentioned in this section

Figure 1: I suggest useing coloured lines. Figure 1b is misleading (acetic acid and ethanol lined after 2 days ).
